# A conserved transcription factor controls gluconeogenesis via distinct targets in hypersaline-adapted archaea with diverse metabolic capabilities

**Rylee K. Hackley**[1,2], **Angie Vreugdenhil-Hayslette**[1¤], **Cynthia L. Darnell**[1], **Amy K. Schmid**[1,2,3]*

**1** Department of Biology, Duke University, Durham, North Carolina, United States of America, **2** University Program in Genetics and Genomics, Duke University, Durham, North Carolina, United States of America, **3** Center for Genomics and Computational Biology, Duke University, Durham, North Carolina, United States of America

¤Current address: Occupational and Environmental Safety Office, Duke University, Durham, NC, United States of America
* amy.schmid@duke.edu

**Data Availability Statement:** Gene expression and transcription factor binding data for this manuscript are accessible in the Gene Expression

## Abstract

Timely regulation of carbon metabolic pathways is essential for cellular processes and to prevent futile cycling of intracellular metabolites. In *Halobacterium salinarum*, a hypersaline adapted archaeon, a sugar-sensing TrmB family protein controls gluconeogenesis and other biosynthetic pathways. Notably, *Hbt. salinarum* does not utilize carbohydrates for energy, uncommon among Haloarchaea. We characterized a TrmB-family transcriptional regulator in a saccharolytic generalist, *Haloarcula hispanica*, to investigate whether the targets and function of TrmB, or its regulon, is conserved in related species with distinct metabolic capabilities. In *Har. hispanica*, TrmB binds to 15 sites in the genome and induces the expression of genes primarily involved in gluconeogenesis and tryptophan biosynthesis. An important regulatory control point in *Hbt. salinarum*, activation of *ppsA* and repression of *pykA*, is absent in *Har. hispanica*. Contrary to its role in *Hbt. salinarum* and saccharolytic hyperthermophiles, TrmB does not act as a global regulator: it does not directly repress the expression of glycolytic enzymes, peripheral pathways such as cofactor biosynthesis, or catabolism of other carbon sources in *Har. hispanica*. Cumulatively, these findings suggest rewiring of the TrmB regulon alongside metabolic network evolution in Haloarchaea.

## Author summary

The breakdown of and synthesis of carbohydrates are central processes for life. The appropriate response to available nutrients is therefore critical for growth and survival. While these metabolic pathways have been studied for decades, it remains unclear the extent to which regulation of these processes is conserved. In the current study, we investigate the regulation and conservation of central metabolism across related species of hypersaline

Omnibus database under the accession GSE227034. DNA sequencing data are available in the Sequence Read Archive at PRJNA947196. All code and input data can be found at https://github.com/hackkr/Har_hispanica_TrmB.

**Funding:** This material is based upon work supported by the National Science Foundation under Grant numbers 1651117, 1615685, and 1936024 to AKS. Any opinions, findings, and conclusions or recommendations expressed in this material are those of the author(s) and do not necessarily reflect the views of the National Science Foundation. The funders had no role in study design, data collection and analysis, decision to publish, or preparation of the manuscript.

**Competing interests:** The authors have no competing interests.

adapted archaea, microorganisms living at nearly saturated salt. Nutrients are intermittently available in hypersaline environments and therefore the appropriate timing of uptake and use or synthesis of carbohydrates is key for growth and survival. We report strong evolutionary conservation of an important regulatory protein and its respective DNA binding sequence across archaeal species. However, the genes controlled by this regulator differ between species in concordance with variation in metabolic capacities between species.

## Introduction

Regulation of glycolytic and gluconeogenic activities in the cell is critical to generate energy and direct carbon flux in the face of variable environments and nutrient availability. In bacteria and eukaryotes, allosteric regulation plays an important role, though regulation at the transcriptional and post-transcriptional levels also occurs [1–4]. However, studies in archaea suggest that allosteric regulation of enzymes involved in central carbon metabolism is less prevalent (reviewed in Ref. [5]). For example, reactions catalyzed by the antagonistic enzyme couples phosphofructokinase (*pfk*) and fructose-1,6-bisphosphatase (*fbp*) are not allosterically regulated in most characterized archaeal enzymes, as they are in bacteria. Archaeal pyruvate kinases appear to be sensitive to allosteric activation by novel ligands AMP and 3-phosphoglycerate [6–8]. Instead of allosteric regulation, studies comparing glycolytic and gluconeogenic growth conditions in archaea indicate that regulation at the transcriptional level is important [9–16].

In *Pyrococcus furiosus* and *Themococcus kodakarensis*, both hyperthermophilic members of Euryarchaea, a conserved transcription factor (TF) controls gluconeogenesis. TrmBL1 in *Pyr. furiosus* and Tgr in *Tcc. kodakarensis* are TrmB family regulators that bind DNA in the absence of glucose to induce the expression of gluconeogenic genes and suppress the expression of glycolytic genes. Both homologs recognize a conserved cis-regulatory motif that is absent from closely related, non-carbohydrate utilizing species [17–19]. The direction of regulation is determined by the motif location relative to other promoter elements: binding downstream of the TATA-box inhibits RNA polymerase recruitment, whereas upstream binding activates transcription.

Though this class of sugar-sensing TrmB regulators is broadly conserved in archaeal and bacterial lineages [20, 21], the majority of homologs are found in Halobacteria, a class of hypersaline-adapted Euryarchaea [22]. The non-carbohydrate utilizing, or nonsaccharolytic, species *Halobacterium salinarum NRC-1* encodes a single TrmB homolog that has been previously characterized. In *Hbt. salinarum*, TrmB regulates more than 100 genes via the same mode of regulation proposed for hyperthermophiles. TrmB regulates the expression of genes in the absence of glucose to activate gluconeogenesis and suppress other metabolic pathways such as amino acid metabolism, cobalamin biosynthesis, and purine biosynthesis [10, 23]. Although glucose is not actively transported into the cell nor catabolized via glycolysis, it is essential for glycosylation of the proteinaceous cell surface layer (s-layer) [24–28]. TrmB plays an essential role in this process by controlling the availability of glucose moieties and expression of enzymes for amino acid metabolism and co-factor biosynthesis, and decreasing flux through purine biosynthetic pathways [10, 23, 29, 30]. The genes encoding phosphoenolpyruvate synthase (*ppsA*) and pyruvate kinase (*pyk*) comprise an important regulatory point controlling carbon flux in *Hbt. salinarum*. Within five minutes of glucose addition, *ppsA* is de-activated and *pykA* is de-repressed by TrmB [23]. Furthermore, TrmB regulates the expression of several other transcription factors, endowing the gene regulatory network (GRN) with

dynamical properties such as transient "just-in-time" expression of metabolic genes during rapid nutrient shifts, suggesting that TrmB is a hub at the core of a larger GRN.

However, the majority of characterized Halobacteria, hereon referred to as haloarchaea for clarity, are carbohydrate utilizers, or saccharolytic. These organisms primarily use the semi-phosphorylative Entner-Doudoroff (spED) pathway for glycolysis, while a modified Embden Meyerhof Parnas (EMP) pathway is predominately utilized during gluconeogenesis and fructose degradation (Fig 1) [31–33]. Glycolysis in saccharolytic haloarchaea has two

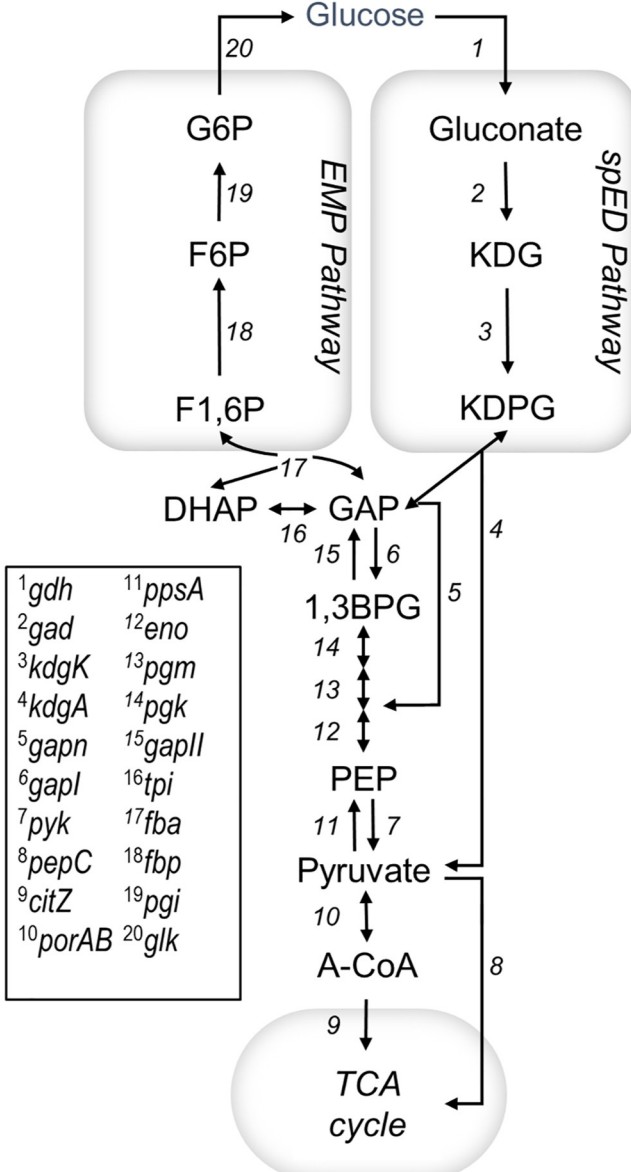

**Fig 1. Central carbon metabolism in Haloarchaea.** In saccharolytic species, glucose is catabolized via the semi-phosphorylative Entner-Doudoroff (spED) pathway (genes 1–10) [33]. In particular, saccharolytic species encode *gapI* (6), while nonsaccharolytic species possess *gapn (5)*. All species are predicted to synthesize glucose via the reverse EMP pathway, or gluconeogenesis (genes 10–20) [5]. Abbreviations are as follows: G6P, glucose-6-phosphate; F6P, fructose-6-phosphate; F1,6P, fructose-1,6-bisphosphate; DHAP, dihydroxyacetone phosphate; GAP, glyceraldehyde-3-phosphate; 1,3BPG, 1,3-bisphosphoglycerate; PEP, phospho*enol*pyruvate; A-CoA, acetyl-CoA; TCA, tricarboxylic acid; KDG, 2-keto-3-deoxygluconate; KDPG, 2-keto-3-deoxy-6-phosphogluconate.

distinguishing features from other archaeal groups that enable the same ATP yield as the spED pathway in Bacteria: (i) a bacterial-type catabolic glyceraldehyde-3-phosphate dehydrogenase (*gapI*), and (ii) an amphibolic archaeal phosphoglycerate kinase (*pgk*) [34]. Nonsaccharolytic species, such as *Hbt. salinarum* and *Natrinema sp.* strain J7–2, possess all genes necessary for a functional spED pathway except that they lack the *gapI/pgk* operon present in saccharolytic genomes and instead encode an archaeal-specific nonphosphorylating glyceraldehyde-3-phosphate (GAP) dehydrogenase (*gapn*) [35–37]. Despite the presence of all necessary genes in the genome, the activity of spED enzymes has not been detected in *Hbt. salinarum* [24, 27, 38]. In particular, because the GAPN reaction proceeds without the formation of 1,3-bisphosphoglycerate coupled to substrate-level phosphorylation, the theoretical ATP yield of glycolysis in nonsaccharolytic haloarchaea is lower than in haloarchaea encoding *gapI*.

As in the other bacteria and eukaryotes, the synthesis of glucose from pyruvate, or gluconeogenesis, proceeds through the reverse EMP pathway in haloarchaea. Enolase (*eno*), phosphoglycerate mutase (*pgm*), triose phosphate isomerase (*tpi*), and phosphoglucose isomerase (*pgi*) enzymes are bidirectional and act in both glycolysis and gluconeogenesis (Fig 1). In contrast, the irreversible enzymes *pgk*, archaeal-specific *gapn*, and *pyk* are specific to glycolytic reactions. The irreversible gluconeogenic enzyme *ppsA* catalyzes the reverse (gluconeogenic) reaction, opposing *pyk* as in bacteria. Similarly, the *gapn* or *gapI* reactions can be reversed by the archaeal class of gluconeogenesis specific phosphorylating glyceraldehyde-3-phosphate dehydrogenase (GAPDH, encoded by *gapII*) [34]. In most archaea except for haloarchaea, dihydroxyacetone phosphate (DHAP) and GAP are converted directly to fructose-6-phosphate by a bifunctional fructose bisphosphate aldolase/phosphatase [5]. In haloarchaea, like in bacteria and eukaryotes, these steps are catalyzed by fructose-bisphosphate aldolase (*fba*) and fructose bisphosphatase (*fbp*) [39].

The role of TrmB has yet to be investigated in other haloarchaea, particularly in models that catabolize diverse sets of carbohydrates and are of interest for industrial applications [40, 41], such as *Haloarcula hispanica* ATCC33960. *Har. hispanica* is a moderately halophilic archaeon that grows on a wide array of carbon sources such as pentose and hexose sugars, disaccharides, three-carbon molecules, and acetate [42, 43]. Moreover, when oxygen, nitrogen, or phosphorus is limited and carbon is abundant, *Har. hispanica* accumulates large quantities of poly (3-hydroxybutyrate-co-3-hydroxyvalerate), or PHBV, a biodegradable plastic alternative [44–46]. Due to its biotechnological potential, metabolic studies thus far have focused on PHBV synthesis [13, 47, 48]. Correct activation of PHBV production implies sensing carbon availability, down-regulation of glycolysis, and activation of the PHBV synthesis pathway. Our understanding of how various signals are integrated to coordinate such a response is still unclear. A better understanding of metabolic regulation generally in *Har. hispanica* is needed to elucidate the function of genes and enzymes for biotechnological applications.

In this study, we characterized TrmB in *Haloarcula hispanica* using high-throughput phenotyping, genome-wide binding assays, and expression experiments to compare its targets with those previously reported in *Hbt. salinarum* [10]. We find that TrmB is essential for growth in gluconeogenic conditions in both species, but when TrmB is deleted in *Har. hispanica* growth can be restored by supplementing a wider variety of carbon sources, in line with its saccharolytic capabilities. We identified 15 robust TrmB binding sites across the genome corresponding to the differential expression of 9 genes predominantly involved in gluconeogenesis. A point of bidirectional regulation by TrmB of the EMP pathway in *Hbt. salinarum* (activation of *ppsA* and repression of *pykA*) is absent in *Har. hispanica*. Instead, we propose that TrmB-dependent induction of the archaeal, gluconeogenic-specific GAPDH is necessary for gluconeogenesis. Together these results suggest an ancestral role for TrmB in enabling gluconeogenesis across

Euryarchaea and highlight this family of transcriptional regulators as an important indicator of metabolic versatility in hypersaline-adapted archaea.

## Materials and methods

### Media & growth conditions

All *Har. hispanica* strains were derived from *Haloarcula hispanica* ATCC33960 type strain. *Har. hispanica* was routinely grown on rich medium for *Haloferax volcanii* modified to contain 23% basal salts (YPC23) supplemented with 0.1% glucose (w/v) [49]. During glucose limitation experiments, strains were grown in casamino acid media modified to 23% basal salt concentration (Hh-CA). Briefly, 23% basal salts contain, per liter, 184 g of NaCl, 34.5 g of $MgSO_4 \cdot 7H_2O$, 23 g of $MgCl \cdot 6 H_2O$, 5.4 g of KCl, and 15.3 mM Tris HCl (pH 7.5). If glucose was supplemented, it was added to a final concentration of 0.1% (w / v) unless otherwise noted. All media were supplemented with uracil (50 $\mu$g/ml) to complement the biosynthetic auxotrophy of the $\Delta$ *pyrF* parent strain unless specified. Other ingredients and media preparation are in line with Allers *et. al.* [49]. All plates were incubated at 37˚C for 8–10 days for single colonies and liquid cultures were cultivated aerobically at 37˚C with 250 rpm orbital agitation.

### Strains, plasmids, & primers

Deletion and integration plasmids were constructed using isothermal assembly [50]. For growth complementation assays, a heterologous expression vector for *Har. hispanica* was constructed from pWL502, a pyrF-based expression vector for *Haloferax mediterranei* [51]. Briefly, a mevinolin resistance cassette was amplified from pNBKO7 and replaced the *Hfx. mediterranei pyrF* at the SmaI and BamHI sites to yield pAKS83. All plasmid sequences were confirmed via Sanger sequencing and propagated in *Escherichia coli* NEB5$\alpha$. Strains, plasmids, and primers used are presented in S1, S2 and S3 Tables, respectively. Gene deletions and chromosomal integrations were performed using two-stage selection and counterselection as described previously [45]. Strains were generated using the spheroplasting transformation method and plated on Hh-CA without supplemental uracil [52]. Resulting colonies were inoculated in 5 ml of YPC23 + glucose and grown for 48 hours and then plated onto Hh-CA + 150 $\mu$g/ml 5-Fluoroorotic acid (5-FOA) + glucose for counterselection.

To verify the complete deletion of all copies of *trmB* in the genome and to check for secondary mutations, genomic DNA was extracted using phenol:chloroform:isoamyl alcohol (25:24:1) followed by ethanol precipitation. TruSeq libraries were prepared and sequenced on Illumina MiSeq by the Center for Genomic and Computational Biology at Duke University (Durham, NC). Reads were aligned to the *Har. hispanica* ATCC33960 genome (GCF_000223905.1, assembly ID ASM22390v1, accessed 2022-10-19) using breseq with default options [53].

### Sequence analysis of *Har. hispanica* TrmB homologs

To identify sugar-sensing TrmB homologs, reference proteomes were searched for protein sequences similar to TrmB$_{Hbt}$ and results were filtered to include hits with identical domain architecture. Domain identities and confidence values were confirmed using hmmscan on the Pfam database [54, 55]. *Har. hispanica* TrmB protein and gene sequences were accessed from NCBI and globally aligned to *Halobacterium salinarum NRC-1* VNG1451C using EMBOSS Needle [56].

## Quantitative phenotyping

Strains were streaked from freezer stocks for each experiment and incubated for 10 days. Single colonies were inoculated in 5 mL YPC23 + glucose and grown to stationary phase ($OD_{600}$ $\sim$ 4). Stationary precultures were then collected, washed twice in Hh-CA and diluted into 200 $\mu$l fresh Hh-CA with or without 25 mM of additional carbon sources to an initial $OD_{600}$ of 0.05. Growth was measured every 30 minutes at 37˚C with continuous shaking for 72–90 hours in Bioscreen C analysis system (Growth Curves USA, Piscataway, NJ). For quantitative analysis, growth curves were blank-adjusted within independent experiments, fitted, and holistic differences across growth phases were summarized using the area under the curve (AUC) [57]. AUC values were averaged across technical replicates, the standard deviation was calculated across biological replicates, and significant differences were evaluated by a two-tailed paired Student's t-test.

## Isolation of uracil prototrophic AKS133 strains

Colonies used for AKS133 RNA-seq experiment that showed *pyrF* expression were subjected to a quantitative growth assay in Hh-CA media ± uracil ± glucose to test for growth in the absence of uracil. All 10 wells containing AKS133 grew in the absence of uracil. These cultures were inoculated into Hh-CA—uracil + glucose for 24 hours and then plated in medium lacking uracil for individual colonies. Of the ten cultures, single colonies were isolated from 7, from which genomic DNA was extracted. Amplification of *pyrF* from genomic DNA confirmed that the endogenous deletion was intact in all prototrophic isolates, ruling out gene conversion, and revealed *pyrF* sequence elsewhere in the genome (primer sequences are listed in S3 Table). We were unable to locate *pyrF* with multiple attempts at arbitrary PCR [58], but note that there is substantial sequence homology between the vector and the *Har. hispanica* genome, particularly near the origins of replication.

## AKS319 strain construction & verification

To construct AKS319, transformations were carried out as described above, except that all media and plates were supplemented with glucose to alleviate selection. Colonies harboring the deletion vector were passaged twice in YPC23 + glucose (for a total of 96 hours), and then exposed to a higher concentration (250 $\mu$g/ml) of 5-FOA to select against *pyrF* and vector retention. After initial confirmation of genotype by PCR, prior to storage and WGS, 5-FOA-resistant colonies were screened for uracil prototrophic growth for 90 hours in Hh-CA media ± uracil ± glucose. Genomic DNA was extracted and WGS was carried out to confirm no reads mapped to the *pyrF* or *trmB* loci.

## Chromatin immunoprecipitation

Strains DF60 and AKS155 were streaked from freezer stocks onto YPC23 plates and incubated for 8 days. Independent colonies were used to start two cultures of DF60 and four cultures of AKS155 in 10 mL YPC23. Precultures were grown for 29 hours ($OD_{600}$ $\sim$ 2) then collected at 6000 rpm for 2 minutes and washed twice with Hh-CA. AKS155 pellets were resuspended in Hh-CA or Hh-CA + glucose and grown to midlog phase (average $OD_{600}$ $\sim$ 0.34) before cross-linking (S1 File). Samples were processed as described by Wilbanks et al. using anti-HA polyclonal antibody (Abcam catalog #ab9110) to immunoprecipitate cross-linked fragments [59]. Libraries were constructed by the Center for Genomic and Computational Biology at Duke University (Durham, NC) using KAPA Hyper Prep kit and Illumina TruSeq adapters, and the 50 base pair, single-ended libraries were sequenced on an Illumina HiSeq 4000.

## ChIP-seq read processing & peak calling

Raw FastQ files were trimmed of adapter sequences with Trim_galore! 0.4.3 and Cutadapt 2.3 and read quality was checked with FastQC 0.11.7. Reads were aligned to the *Har. hispanica* genome with Bowtie2 2.3.4.3 [60]. Aligned sequence files were then sorted, indexed, and converted to binary format with samtools 1.9 [61]. Before calling the peaks, the fragment length was optimized for each IP and input control sample using ChIPQC 1.30 [62]. Binding peaks were called from sorted bam files using Mosaics 2.32 in R 4.1.2 with the calculated fragment lengths and FDR < 0.01. DiffBind 3.4.11 was used to merge peaks that were shared in at least 3 biological replicates for samples grown without glucose (N = 4) and all samples grown in the presence of glucose (N = 2) [63]. Briefly, binding peaks were merged if they overlapped by at least one base pair, and consensus peaks were then trimmed to 300 base pair width centered around the consensus peak maximum. Samples were RLE normalized. Separately, we used DiffBind to identify peaks shared between glucose-replete and depleted samples that exhibited differential binding. Peaks were visually verified using the trackvieweR package to determine an enrichment cutoff [64]. Then, consensus peaks were annotated using IRanges and GenomicFeatures to identify genomic features overlapping and adjacent to TrmB binding peaks [65]. For the per base enrichment plot, bam files were extended to the estimated average fragment size by ChIPQC (150 base pairs), converted to BEDGRAPH format, and scaled according to sequencing depth before calculating the enrichment ratio and averaging across replicates as described by Grünberger *et al.* [66]. The fifteen regions identified via peak-calling exactly correspond with regions surpassing log2 fold enrichment of 4.

## RNA-isolation & sequencing

Strains DF60 and ASK133 or AKS319 were streaked from freezer stocks onto YPC23 + 0.1% glucose plates and incubated for 10 days. Four single colonies of DF60 and ASK133 or AKS319 were inoculated in 5 mL YPC23 medium with 0.1% glucose and grown aerobically to stationary phase ($OD_{600} \sim 4$). Cells were washed twice with Hh-CA medium without glucose and transferred into fresh 50 mL Hh-CA with or without 0.1% glucose at an initial density of $OD_{600} \sim 0.4$. After 24 hours, cultures were harvested and RNA extracted using Absolutely RNA Miniprep kit (Agilent Technologies, Santa Clara, CA) followed by additional DNAse treatment with Turbo DNAse (Invitrogen, Waltham, MA). Total RNA was quantified using Agilent Bioanalyzer RNA Nano 6000 chip (Agilent Technologies, Santa Clara, CA). The absence of DNA contamination was determined on 200 ng of RNA in 25 PCR cycles. Ribosomal RNA was removed with the PanArchaea riboPOOL kit according to the manufacturer protocol (siTOOLs Biotech, Germany), and sequencing libraries were constructed with NEBNext UltraII Directional RNA Library Preparation Kit (Illumina, #E7760) as described previously [67]. The fragment size of the libraries was measured using the Agilent Bioanalyzer DNA 1000 chip and then pooled and sequenced on NovaSeq6000 at the Center for Genomic and Computational Biology at Duke University (Durham, NC).

## Differential expression analysis & clustering

FastQ files were processed as described for ChIP-seq, except alignment by Bowtie2 was done using pair-end mode. After alignment, counts were calculated using featureCounts requiring complete and concordant alignment of paired reads [68]. Samples were considered outliers and removed prior to differential expression analysis if they had a pairwise correlation lower than R = 0.6 with all other replicates. Differential expression analysis was carried out in DeSeq2 with log2 fold-change and FDR cutoffs of 1 and 0.05, respectively, using an experimental design informed by TrmB model of regulation in *Hbt. salinarum* [10, 69]. To cluster

differentially expressed genes based on expression patterns across strain and condition, normalized counts were mean and variance standardized and subjected to k-means clustering and visualized as described in [70].

### Functional enrichment

Annotated and predicted gene functions were accessed from EggNOG 5.1 [71]. Genes adjacent to TrmB binding sites were then tested for functional enrichment relative to the whole genome using a hypergeometric test with Benjamini and Hochberg multiple testing correction.

### Motif discovery & scanning

TrmB binding motif was identified using the command line version of MEME Suite 5.5.1. Specifically, sequences under consensus peaks (300 base pairs) were extracted and analyzed with XSTREME for motif discovery, motif enrichment, and motif comparison [72]. Default settings were used except that the maximum motif width was set to 19 base pairs and a first-order Markov background model was generated using non-coding regions of the *Har. hispanica* genome to control for dinucleotide frequencies. The recognition motif reported for *Hbt. salinarum* was included for downstream motif enrichment and similarity assessment (see reference [10] supplemental table 5). The motif identified was robust to zero and second-order background models. To identify motif occurrences genome-wide, the *Haloarcula hispanica* reference genome was scanned using FIMO with a p-value cutoff of $1 \times 10^{-4}$.

## Results

### Four TrmB homologs are encoded in the *Har. hispanica* genome

Using bioinformatic analysis, we investigated the conservation of TrmB. We detected 619 proteins with identical domain architecture as *Hbt. salinarum* TrmB across archaeal and bacterial Pfam reference proteomes (S1 File). Of these, almost half were encoded in haloarchaeal genomes (Fig 2A). On average, Haloarchaea genomes (n = 97) contain three TrmB homologs per genome, compared to an average of one homolog outside of haloarchaea (n = 238), indicating lineage-specific expansion of this class of regulators (p-value $< 2.2 \times 10^{-16}$, one-sided Wilcoxon test). Some regulators containing the TrmB DNA-binding domain have been recently shown to function during oxidative stress in haloarchaea [22], but proteins with both a TrmB DNA-binding domain and carbohydrate-binding domain have not been functionally characterized in haloarchaea aside from VNG_1451C in *Hbt. salinarum* (hereafter referred to as TrmB$_{Hbt}$).

To determine the most likely functional ortholog of *Hbt. salinarum* TrmB (VNG_1451C) in the *Har. hispanica* genome, we searched the 148 proteins with predicted DNA-binding capabilities. Ten of these putative transcription factors have a TrmB DNA-binding domain and four contain both the TrmB DNA-binding domain and sugar-binding domain (PF01978 and PF11495, respectively): HAH_0923, HAH_1548, HAH_1557, and HAH_2795 (Fig 2B). Multiple sequence alignment revealed that HAH_0923, HAH_1548, and HAH_2795 maintain two of the six active site residues required for sugar-binding in *Thermococcus litoralis*, while HAH_1557 preserved four of the six residues [73]. Both G320 and E326 residues, which when mutated in *Tcc. litoralis* drastically reduce the binding affinity for sucrose and maltose, are conserved across the haloarchaeal TrmB proteins analyzed [73]. In contrast, N305, which is specific for maltose but not sucrose binding, is not conserved outside of hyperthermophiles [74]. Of the TrmB homologs in *Har. hispanica*, HAH_1548 exhibits the highest amino acid

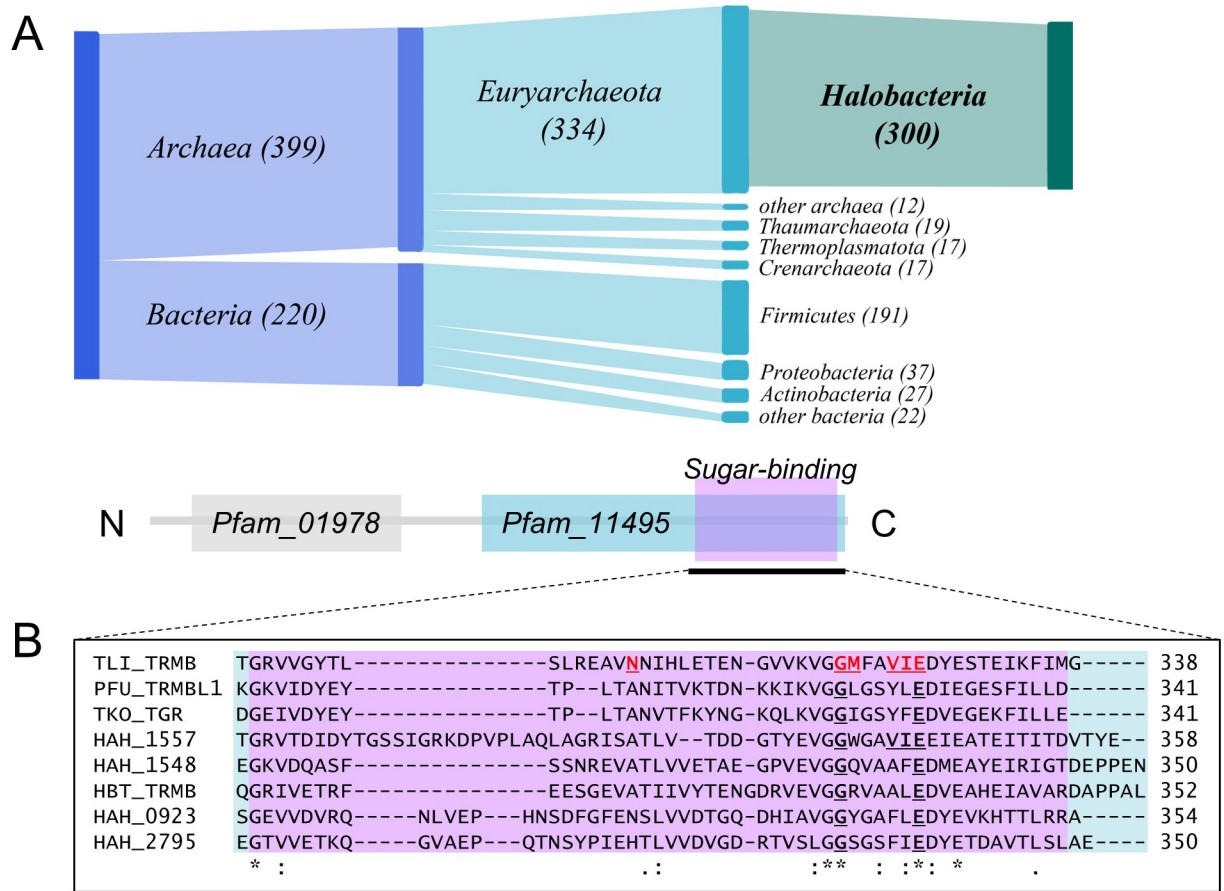

**Fig 2. Sugar-sensing TrmB homologs.** A) Phylogenetic distribution of TrmB proteins that also contain a carbohydrate-binding domain. B) Multiple sequence alignment of the sugar-binding domain of *Har. hispanica* TrmB homologs. Red, underlined residues are essential for sugar binding in *Tcc. litoralis*, conserved residues are in bold underline [73, 74]. Asterisks below the alignment denote identical residues and dots represent similar residues. Organism abbreviations and locus tags are as follows: TLI_TRMB, *Tcc. litoralis* OCC_03542; PFU_TRMBL1, *Pyr. furiosus* PF0124; TKO_TGR, *Tcc. kodakaraensis* TK1769; HBT_TRMB, *Hbt. salinarum* VNG_1451C. *Har. hispanica* TrmB homologs are named according to their identity to characterized proteins.

sequence identity to TrmB$_{Hbt}$, and best agreement with the sugar-binding domain consensus (e-value = $3.3 \times 10^{-20}$, Table 1).

In *Tcc. litoralis* and *Tcc. kodakarensis*, the *trmB* homolog is located near genes that encode carbohydrate ABC transporters. However, in *Hbt. salinarum* and *Pyr. furiosus*, these genes are

**Table 1.** *Har. hispanica* TrmB homologs.

| TrmB homolog | gene name | protein identity | protein positives | DBD e-value | SBD e-value |
|---|---|---|---|---|---|
| VNG_1451 | *trmB* | 100% | 100% | $1.8 \times 10^{-70}$ | $6.1 \times 10^{-20}$ |
| **HAH_1548** | *trmB* | **56.5%** | **74.2%** | $1.1 \times 10^{-68}$ | $3.3 \times 10^{-20}$ |
| HAH_2795 | *trmB2* | 31.2% | 47.5% | $1.5 \times 10^{-70}$ | $3.5 \times 10^{-14}$ |
| HAH_1557 | *trmB3* | 27.2% | 48.5% | $2.9 \times 10^{-78}$ | $1.2 \times 10^{-14}$ |
| HAH_0923 | *trmB4* | 25.4% | 39.9% | $1.9 \times 10^{-60}$ | $4.0 \times 10^{-17}$ |

Protein sequences were globally aligned to *Hbt. salinarum* TrmB. DNA-binding domain (DBD) and sugar-binding domain (SBD) confidence scores were calculated with hmmscan [54].

absent from the genomic region surrounding *trmB* [10, 75]. Genomic context analysis revealed that HAH_1548 is upstream of a putative carbohydrate ABC transporter and glucose-1-dehydrogenase. HAH_1557 flanks the same ABC transport system but has a substantially lower identity to TrmB_Hbt (Fig 2B, Table 1). Thus, HAH_1548 was distinguished via sequence alignment and genomic context as the likely functional homolog of TrmB_Hbt (for clarity, hereafter we refer to the HAH_1548 gene as *trmB_Har* and the translated protein as TrmB_Har).

## TrmB_Har is essential for growth in gluconeogenic conditions

To assess whether TrmB_Har plays a physiological role in carbohydrate metabolism and gluconeogenesis, we deleted *trmB_Har*. Because haloarchaea are polyploid [76], it is possible for copies of the wild-type locus to persist at levels too low to detect via Sanger sequencing [77]. Therefore, the genotypes of all strains in this study were confirmed with short-read whole genome sequencing (WGS). WGS of the *trmB_Har* deletion strain confirmed it to be free of second-site mutations and chromosomal copies of *trmB_Har* (S1 Fig, SRA accession PRJNA947196). Notably, DF60, the *Δ pyrF* parent strain, harbors a previously unreported 2.6 kb deletion disrupting HAH_2675–79, encoding *flgA1* and *cheW1*. This region is absent in all strains derived from the *Δ pyrF* strain in this study.

We subjected both *Δ trmB_Hbt* and *Δ trmB_Har* to a panel of nine ecologically and physiologically relevant carbon sources at an equimolar concentration to facilitate comparisons between species (Fig 3). As expected from previous reports, the *Hbt. salinarum Δ trmB* strain exhibits a severe growth defect under gluconeogenic conditions [10]. Only glucose and glycerol stimulate growth in the deletion strain (Fig 3A). The slight differences between these results and those reported by Schmid *et al.* are due to differences in supplemental carbon concentration (389mM and 760mM for glucose and glycerol, respectively, [10]).

Growth assays also revealed a severe growth defect of the *Δ trmB_Har* strain in Hh-CA (Fig 3B). Without additional carbon sources, amino acids are the sole nitrogen and carbon source in Hh-CA, so cells rely on gluconeogenesis to synthesize all required glucose. Normal growth could be restored with heterologous expression of *TrmB_Har* (S2 Fig). In contrast to *Hbt. salinarum*, *Δ trmB_Har* growth could be restored by the addition of glucose, fructose, sucrose, or glycerol. Xylose stimulated growth to 57% of the parent strain, as measured by the area under the growth curve (AUC), but growth remains depressed relative to the parent strain in the presence of ribose, pyruvate, acetate, and galactose. Accelerated growth of the parent strain in media supplemented with pyruvate and ribose contributes to the lower growth ratios reported for those carbon sources (S2 Fig).

We next titrated the concentration of glucose to identify the minimum amount that restored normal growth (Fig 3C). Relative to parent strain growth in the absence of carbon, *Δ trmB_Har* was stimulated 0.31, 0.81, 1.42, 1.73, 1.74-fold with no supplemental carbon source, 0.01%, 0.05%, 0.1%, and 0.5% glucose (w/v), respectively. Glucose concentrations above 0.1% did not further stimulate growth, identifying 0.1% glucose, or 5.55 mM, as the minimal glucose concentration needed to rescue growth. These data demonstrate that TrmB is indispensable for growth under gluconeogenic conditions in haloarchaea with various metabolic capabilities, and implicate glucose in the function of TrmB in both *Hbt. salinarum* and *Har. hispanica*.

## TrmB_Har binds promoters of genes involved in central carbon metabolism

To determine whether TrmB_Har functions as a transcriptional regulator, we located TrmB_Har-DNA interactions with chromatin immunoprecipitation coupled to sequencing (ChIP-Seq) in the presence and absence of glucose. A *trmB_Har*::hemagglutinin (HA) epitope fusion at the endogenous locus was constructed for these experiments and confirmed via WGS (S1 Fig).

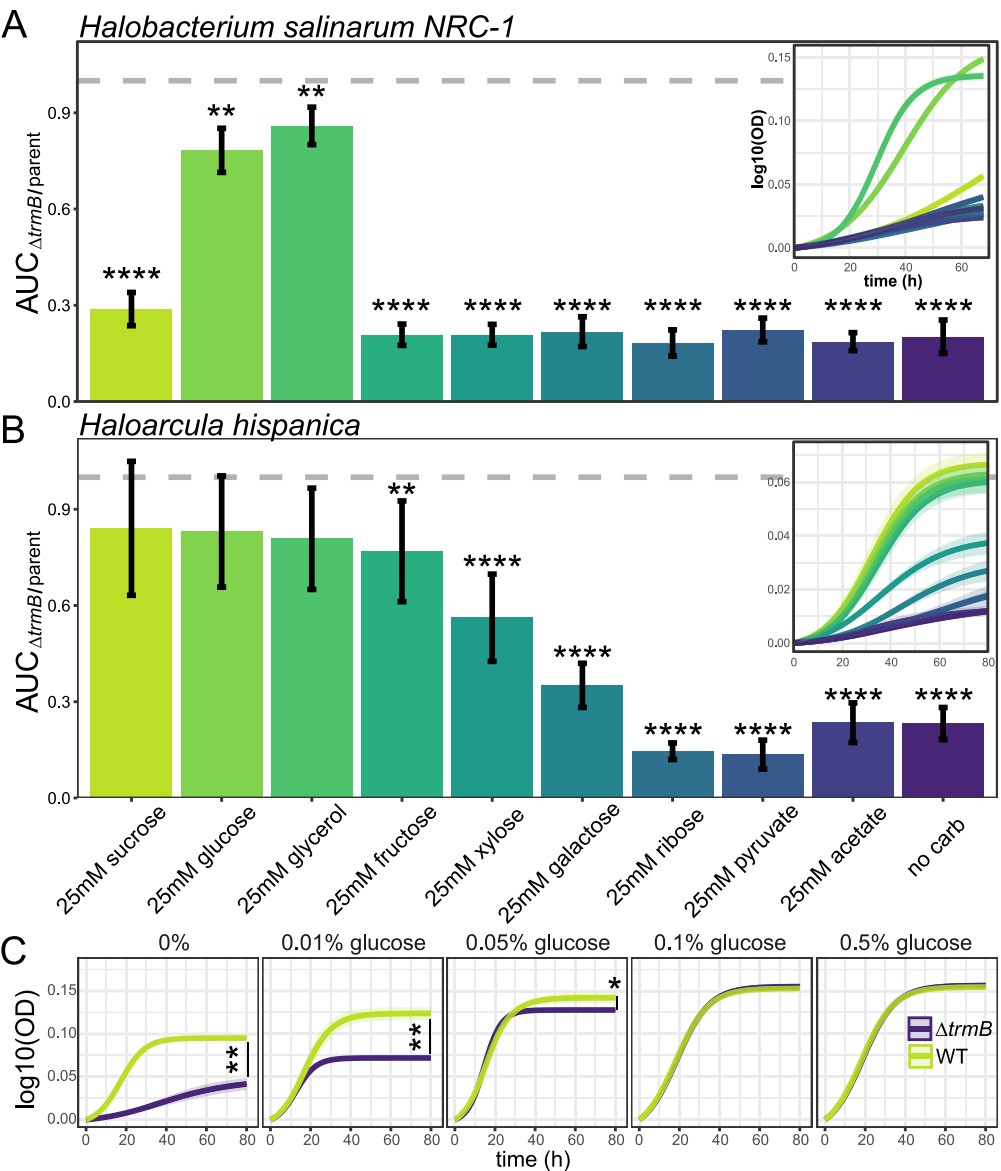

**Fig 3. TrmB$_{Har}$ is essential for gluconeogenic growth.** Proportion of parent growth achieved by (A) $\Delta$ $trmB_{Hbt}$ and (B) $\Delta$ $trmB_{Har}$ with various carbon supplements, as measured by area under the growth curve (AUC). Error bars indicate the standard deviation over a minimum of 3 biological replicates, each in technical triplicate. Insets show the fitted, log-transformed growth curves of $\Delta$ $trmB$ strains in each condition. Asterisks indicate the significance of growth difference relative to the parental strain in each respective condition. C) Effect of increasing glucose concentrations on the growth of *Har. hispanica* $\Delta$ $trmB$ and $\Delta$ $pyrF$ strains. Solid lines represent the mean of the log-transformed growth curve, and shaded regions depict 95% confidence intervals. For all panels, asterisks indicate significance: * *p*-value < 0.05; ** < 0.01; **** < 0.0001.

There was no difference in the growth of the tagged strain from the parent strain in the absence of glucose, suggesting the C-terminal fusion retained full functionality (S2 Fig).

Consistent with the model of regulation reported in *Hbt. salinarum* and homologs in hyper-thermophiles [10, 17, 78], TrmB$_{Har}$ primarily binds DNA when glucose is absent (Fig 4A). Under this condition, fifteen reproducible TrmB$_{Har}$ binding sites, or peaks, were enriched relative to the input control (Table 2, Fig 4B, Methods). Of the 15 peaks, seven exhibited

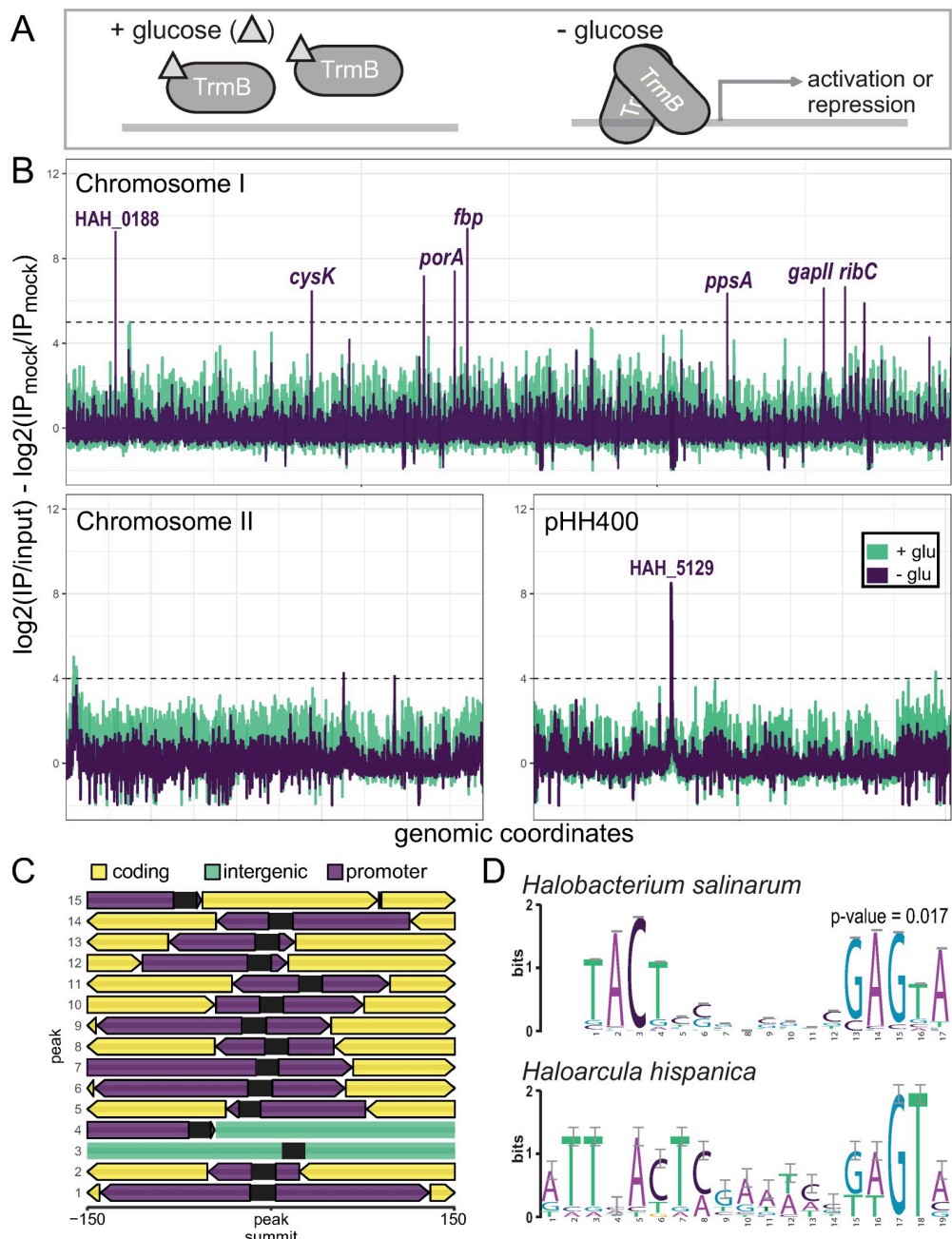

**Fig 4. Genome-wide TrmB$_{Har}$ binding sites.** A) Proposed TrmB mechanism of regulation from *Hbt. salinarum* and *Thermococcus kodakaraensis* [10, 19]. B) Per base enrichment of reads across the genome. Samples were normalized to paired input samples and averaged across replicates. Averages were normalized relative to untagged controls and log-transformed. C) Genomic context of TrmB$_{Har}$ binding sites. Sites were trimmed to 300 base pairs centered around peak summits. Promoter regions are shown in purple, intergenic regions in green, coding regions in yellow, and motif locations in black. Regions are sorted by enrichment, or peak size. D) XSTREME motif analysis of selected regions. The 19-base pair motif identified is similar to the binding motif reported for TrmB$_{Hbt}$ (p-value = 0.017, TomTom [79]).

**Table 2. *Har. hispanica* TrmB binding-sites.**

| peak rank | genomic context | LFC peak | LFC -gluc | motif | location | locus | name | LFC RNA | annotation |
|---|---|---|---|---|---|---|---|---|---|
| 1 | ← * → | 213.66 | 6.94 | yes | promoter<br>promoter | HAH_1418<br>HAH_1419 | *paaH*<br>*fbp* | -6.58 | 3-hydroxybutyryl-CoA dehydrogenase class 1 fructose-bisphosphatase |
| 2 | * → | 154.1 | 6.66 | yes | promoter | HAH_0188 | | -4.63 | hypothetical protein |
| 3 | – * – | 121.07 | | yes | intergenic | | | | |
| 4 | * → | 92.46 | | yes | promoter | HAH_4332 | *gdh* | | glucose dehydrogenase |
| 5 | * → | 85.71 | 5.96 | yes | promoter | HAH_5129 | | 4.72 | sugar porter family MFS transporter |
| 6 | ← * → | 49.84 | 4.46 | yes | promoter<br>promoter | HAH_1365<br>HAH_1366 | *porA* | | flavodoxin reductase family I pyruvate:ferredoxin oxidoreductase |
| 7 | * → | 47.56 | | yes | promoter | HAH_0887 | *cysK* | -1.36 | cysteine synthase A |
| 8 | *→ | 39.81 | 4.69 | yes | promoter | HAH_1264 | | | universal stress protein |
| 9 | ← * → | 37.36 | 5.22 | yes | promoter<br>promoter | HAH_2729<br>HAH_2730 | *gapII* | -4.13 | aminopeptidase type II GAP dehydrogenase |
| 10 | * → | 30.84 | | yes | promoter | HAH_2323 | *ppsA* | | phosphoenolpyruvate synthase |
| 11 | ← * → | 18.73 | 4.65 | yes | promoter<br>promoter | HAH_2805<br>HAH_2806 | *ribC* | -2.89 | hypothetical protein riboflavin synthase |
| 12 | * → | 11.22 | | yes | promoter | HAH_3039 | | | AsnC family transcriptional regulator |
| 13 | ← * → | 8.39 | | yes | promoter<br>promoter | HAH_1011<br>HAH_1012 | | | CBS domain-containing protein gfo/Idh/MocA family oxidoreductase |
| 14 | * → | 6.47 | | yes | promoter | HAH_5130 | | 2.87 | universal stress protein |
| 15 | ← * → | 6.29 | | yes | promoter<br>promoter | HAH_5033<br>HAH_5034 | | -3.30<br>-3.43 | rubrerythrin-like domain protein Glu/Leu/Phe/Val dehydrogenase |

Genomic context indicates the orientation of adjacent genes (* indicates the peak). Peak log-fold change (LFC peak) represents the enrichment of IP samples relative to input controls. LFC -gluc represents the relative enrichment between no glucose and glucose conditions after relative log expression (RLE) normalization in DiffBind. Rows without LFC -gluc values were detected only in samples without glucose, except for HAH_5130 which was detected stably across conditions. The magnitude of TrmB-dependent differential expression as calculated by DeSeq2 is reported in LFC RNA column. Negative values indicate a reduction of glucose-dependent expression when *trmB_{Har}* is deleted relative to the parent strain. Empty cells indicate no significant differential expression was detected.

significantly increased binding in the absence of glucose relative to glucose-replete samples (FDR ≤ 0.1). Seven other peaks were exclusively detected in the absence of glucose, and one was observed in all samples regardless of glucose availability (S1 File, S3 Fig).

TrmB_{Har} binding sites were predominantly located in non-coding regions of the genome (Fig 4C): 41% of the bases in enriched regions were non-coding (p-value $< 1 \times 10^{-15}$, binomial test), despite non-coding sequences comprising only 13% of the *Har. hispanica* genome. In the absence of experimentally characterized promoters for *Har. hispanica* up to 250 base pairs of non-coding sequences upstream of translational start sites were considered promoter regions. TrmB_{Har} binding sites overlap the promoter regions of 20 genes (Table 2). Of those, five are homologous to TrmB_{Hbt} targets by reciprocal protein blast, all in the canonical EMP gluconeogenic pathway: pyruvate oxidoreductase (encoded by *porA/B*), phosphoenolpyruvate synthase (*ppsA*), GAPDHII (*gapII*), and fructose-bisphosphatase (*fbp*). Overall, the predicted function of genes near binding sites were significantly enriched for carbohydrate transport and metabolism (adjusted p-value $< 7.3 \times 10^{-5}$, hypergeometric test). Notably, the second largest peak is located in the promoter of HAH_5129, a sugar major facilitator superfamily (MFS) transporter (PF00083, e-value $< 3.0 \times 10^{-123}$), highlighting this gene as a candidate for the primary glucose transporter in *Har. hispanica*.

Computational *de novo* motif detection using 300 base pairs around binding summits revealed a 19 base pair partially palindromic binding motif (Fig 4D). The motif is similar to

the TrmB recognition sequence reported for *Hbt. salinarum* (p-value = 0.017, calculated with TomTom [79]), and robust to various background models (S4 Fig). The motif occurs 235 times throughout the genome, and of those, 67 instances occur in the promoter regions of 58 genes (motif occurrences on opposite strands were considered distinct, S1 File). Like genes near peaks, genes with motifs located in promoter regions are enriched for functions in the transport and metabolism of carbohydrates and amino acids (hypergeometric test adjusted p-value $< 5.2 \times 10^{-5}$ and 0.003, respectively). Together, these data support the model that in the absence of glucose TrmB$_{Har}$ recognizes a conserved cis-regulatory sequence to bind DNA near genes primarily involved in carbon metabolism.

## Transcriptome profiling suggests cryptic integration of deletion vector, resulting in uracil prototrophy in *Δ trmB$_{Har}$* strains

We conducted transcriptome profiling to determine the direction of TrmB-dependent regulation of genes near binding sites and to elucidate TrmB$_{Har}$ targets in instances where peaks were located over the promoters of divergent genes. Unexpectedly, we observed high *pyrF* expression even though the *pyrF* gene (the *ura5* homolog) is deleted in the parent strain to enable genetic manipulation [45]. Transcripts mapping to the *pyrF* locus were not present in any parent samples but were detected in all *Δ trmB$_{Har}$* (AKS133) samples regardless of glucose availability. Subsequent phenotyping and genotyping confirmed stably inherited uracil prototrophy in 7 of 10 independent AKS133 isolates and ruled out stock contamination, vector propagation, and gene conversion at the endogenous locus (S5 Fig; Methods). This points to homologous recombination of the deletion vector, which harbors *pyrF*, elsewhere in the genome.

   We carefully generated a new *Δ trmB$_{Har}$* strain, AKS319, and repeated the experiment. Despite additional safeguards during strain generation and genotype confirmation by WGS, 4 of 8 *Δ trmB$_{Har}$* samples exhibited high *pyrF* expression (S6 Fig; Methods). Due to this variation, however, we were able to investigate the fitness and transcriptomic consequences of *pyrF* expression in the AKS319 strain and across *Δ trmB$_{Har}$* strains (S6 Fig). AKS319 exhibits a more extreme growth defect relative to AKS133, which can be mostly, but not completely, ameliorated by glucose supplementation. However, expression of *pyrF* has little effect on the overall transcriptome: average counts per gene are highly correlated across all *Δ trmB$_{Har}$* samples regardless of *pyrF* expression both before (S6 Fig) and after batch correction(Fig 5A, S1 File). The repeated and apparently independent instances of vector integration resulting in uracil prototrophy suggests that both deletion strains are likely mixed populations with a small proportion of cells retaining vector sequence. TrmB$_{Har}$ may be conditionally essential in the DF60 background, explaining this phenomenon.

## TrmB$_{Har}$ activates genes involved in gluconeogenesis and tryptophan biosynthesis and represses those encoding glucose uptake

Based on the model of TrmB regulation described in *Hbt. salinarum*, we reasoned that direct targets of TrmB$_{Har}$ would exhibit differential expression only when TrmB is present and active, i.e., in the parent strain grown in the absence of glucose (Fig 4A). This assumption was made explicit in the DeSeq2 model used to identify differentially expressed genes [69]. Using this framework, we detected 32 genes that were significantly differentially expressed (FDR $< 0.05$; LFC $> 1$) in the parent vs mutant strain across two independent analyses, one including and the other excluding samples with counts mapping to *pyrF* (Fig 5B, S1 File). Because *pyrF* expression, when present, is not glucose-dependent, it was not considered significant using this explicit model.

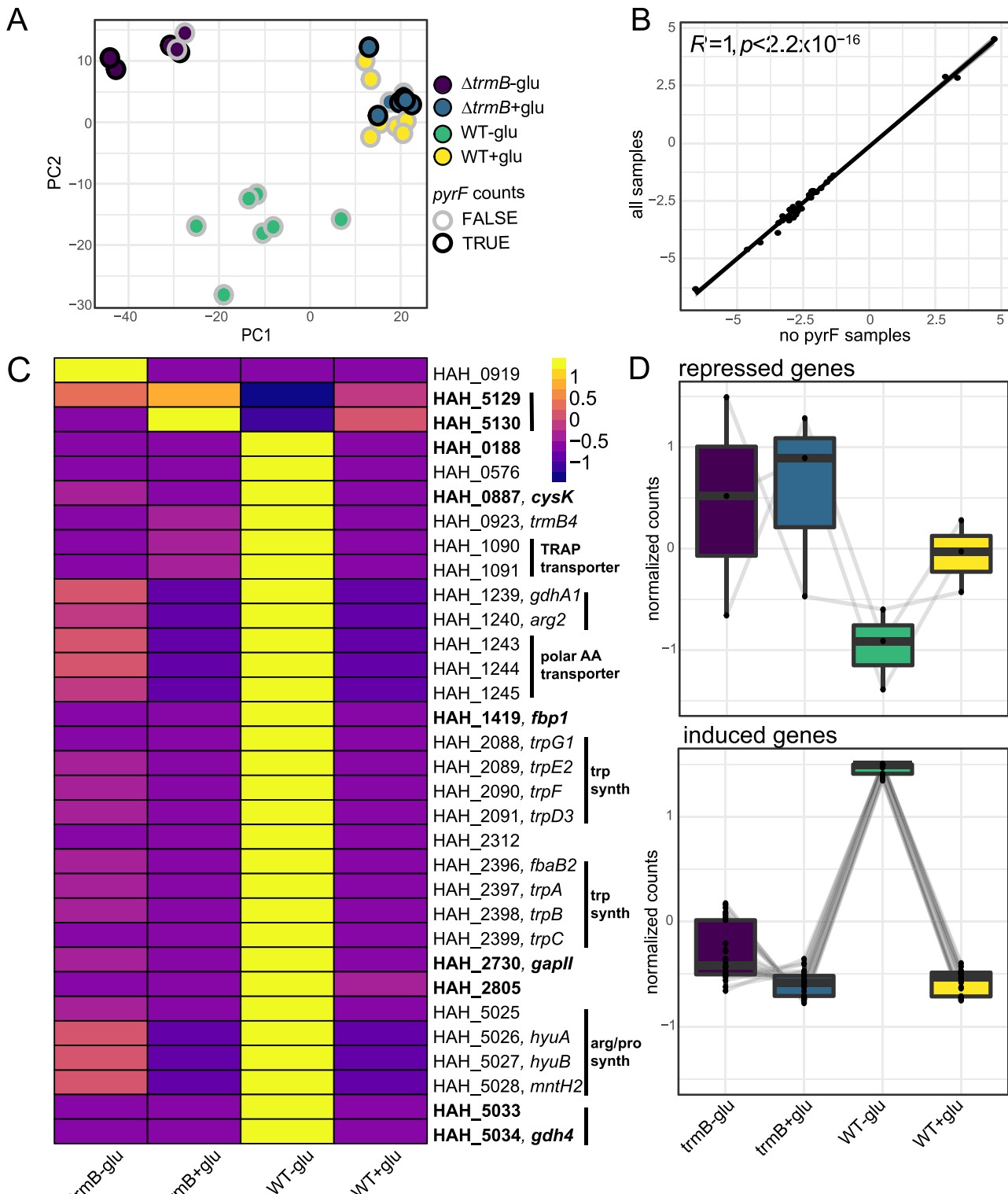

**Fig 5. TrmB$_{Har}$-dependent expression.** A) Principal component analysis of AKS133 and AKS319 samples after batch correction. Samples with transcripts mapping to *pyrF* are outlined in black. B) Correlation of 32 differentially expressed genes from analyses including all *Δ trmB$_{Har}$* samples (y-axis) and only samples with no detected *pyrF* expression (x-axis). C) Normalized expression pattern of 32 genes in (B). Yellow indicates elevated expression and purple indicates reduced expression. Genes near peaks are bolded and genes predicted to be co-transcribed are indicated by a black bar. If available, the predicted function of the operon was provided. D) Three and 29 genes have patterns consistent with TrmB-dependent repression and induction, respectively. Light grey lines connect normalized expression values for each transcript across strain and condition.

Of the 32 genes, three are significantly up-regulated in $\Delta$ $trmB_{Har}$ relative to the parent strain in the absence of glucose: glutamate synthase (encoded by *HAH_0919*), a universal stress protein (*HAH_5129*), and the MFS family sugar transporter (*HAH_5130*) (Fig 5C and 5D). The remaining 29 genes exhibit the opposite pattern: they are significantly down-regulated in the deletion strain, exhibiting a pattern across strains and conditions consistent with transcriptional activation by TrmB. The predicted products of these 29 genes are enriched for functions in amino acid transport and metabolism (adjusted p-value = $1.085 \times 10^{-10}$, hypergeometric test) including two tryptophan biosynthesis operons, cysteine synthase, a putative amino acid transporter, and a TRAP transporter (Fig 5D). Genes whose expression is induced also include those predicted to function in gluconeogenesis (*gapII* and class I fructose-bisphosphate encoded by *fbp*) and two small hypothetical proteins less than 50 amino acids in length (*HAH_0188* and *HAH_2805*).

Integrating across experiments, nine genes are adjacent to $TrmB_{Har}$ binding sites containing a palindromic cis-regulatory motif, and are differentially expressed (Table 2, Fig 4D). Taking the ChIP-seq, RNA-seq, and motif evidence together, we conclude that these genes comprise the high-confidence regulon under the direct transcriptional control of $TrmB_{Har}$ (Fig 5C bolded, Fig 6A). For these targets, we also determined whether the relative TrmB motif distance from the putative TATA box or start codon was predictive of the direction of regulation (S4 Table). The archaeal promoter architecture resembles a simplified version of that found in eukaryotes, including a TATA box located around -26 bp and a BRE element around -33 bp [80, 81]. Generally, motifs were upstream of predicted transcription initiation features, consistent with our observation that TrmB acts primarily as an activator. In contrast, genes repressed by TrmB binding had motifs that overlapped the predicted B recognition elements, though HAH_0188 is a notable exception (S4 Table).

Given the small number of TrmB binding sites across the genome, it was surprising that only half of the binding sites were near genes that exhibited differential expression. We wondered if there might be other regulatory mechanisms at play, namely $TrmB_{Har}$-dependent regulation of small or antisense RNAs, which have recently been described in other haloarchaeal species [82–85]. Since we required correct strand orientation when generating transcript counts, reads mapping to the non-coding strand would not have been considered in the downstream differential expression analysis. We visualized strand-specific reads near peaks and identified an unannotated transcript approximately 200 base pairs in length near the sole intergenic peak in our data (peak 3, Table 2). This transcript is induced when TrmB is active and contains a predicted promoter sequence upstream (S7 Fig). We also saw evidence for a transcript anti-sense to HAH_0887 that appears to be induced when TrmB is active, and that both genes encoding putative small proteins (HAH_0188 and HAH_2805) appear to have extended 3' UTRs (S7 Fig). We did not find any evidence for unannotated transcripts for peaks near HAH_1010, HAH_1264, HAH_3039, or HAH_4332. Unannotated transcripts require validation and further characterization to be considered direct targets of $TrmB_{Har}$, but support the hypothesis that TFs may regulate the expression of regulatory RNAs in haloarchaea, as has been reported for methanogens [86].

## Discussion

Our data facilitate the general comparison of the TrmB regulon in Euryarchaea and more closely between *Hbt. salinarum* and *Har. hispanica*. We detected fewer $TrmB_{Har}$ binding sites than TrmBL1 in *Pyr. furiosus* across the genome (15 and 28, respectively [17]). Moreover, $TrmB_{Har}$ acts primarily as an activator of gluconeogenic genes while TrmBL1 was predicted to repress most of its targets, which were involved in glycolysis (21, based on relative motif

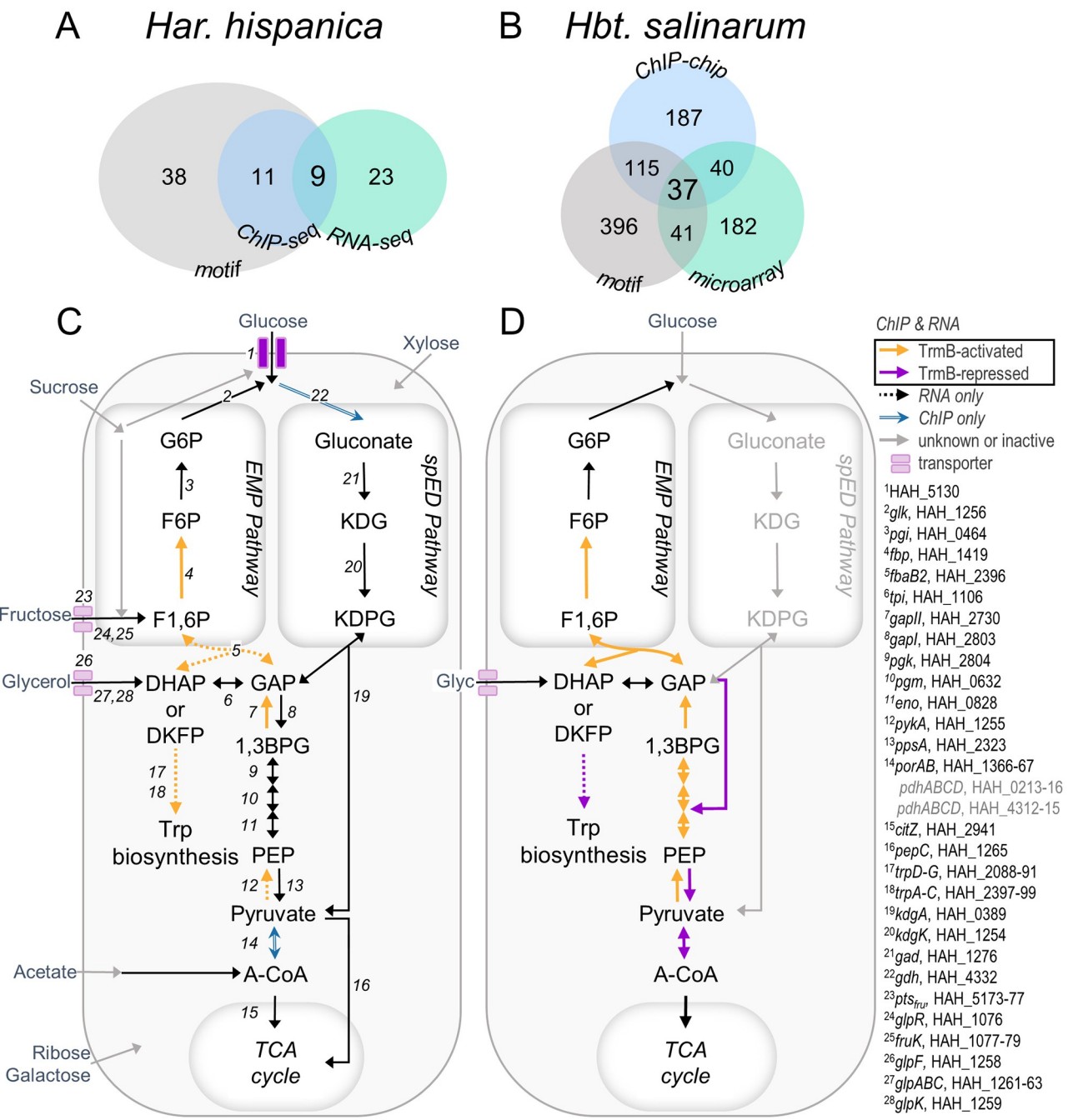

**Fig 6. Comparison of TrmB targets in gluconeogenesis in haloarchaea.** Summary of experimental evidence of TrmB regulon in (A) *Har. hispanica* and (B) *Hbt. salinarum*. Data for *Hbt. salinarum* summarized from figures 4 and 6 of Schmid *et. al.* [10]. C, D: Summary of TrmB targets that encode enzymes in central carbon metabolism for each species. Targets in the high-confidence regulon are indicated with solid, colored arrows. Genes differentially expressed but lacking a nearby binding site are indicated with dashed arrows. Targets that were near binding sites but not differentially expressed are indicated in light blue. Solid gray arrows indicate that specific carbohydrate transport systems are unknown or that genes are present but no enzymatic activity has been detected. Genes predicted to encode the necessary enzymes are labeled (see S8 Fig for normalized expression values). Additional putative pyruvate oxidation systems that are not regulated by TrmB are listed in gray. G6P, glucose-6-phosphate; F6P, fructose-6-phosphate; F1,6P, fructose-1,6-bisphosphate; DHAP, dihydroxyacetone phosphate; DKFP, 6-deoxy-5-ketofructose-1-phosphate; GAP, glyceraldehyde-3-phosphate; 1,3BPG, 1,3-bisphosphoglycerate; PEP, phosphoenolpyruvate; A-CoA, acetyl coenzyme A; KDG, 2-keto-3-deoxygluconate; KDPG, 2-keto-3-deoxy-6-phosphogluconate.

location) [17]. Similarly to TrmBL1, however, our *in vivo* data suggest that expression of TrmB$_{Har}$ is not autoregulated, contrary to reports of other homologs *in vitro* [10, 87]. Conserved regulon members include *gapII* and an MFS transporter, which are activated and repressed, respectively.

At the sequence level, only two homologs are near TrmB binding sites and exhibit congruent differential expression patterns in both haloarchaeal species: *fbp* and *gapII* (Fig 6C and 6D). In general, much more of the triose portion of the EMP pathway (sometimes referred to as "lower glycolysis", although here "upper gluconeogenesis" would be more appropriate) appears to be regulated by TrmB in *Hbt. salinarum* relative to *Har. hispanica* (Fig 6). For example, TrmB represses and activates genes that encode pyruvate kinase (*pyk*) and phosphoenolpyruvate synthase (*ppsA*), respectively, in *Hbt. salinarum*. In *Har. hispanica*, this control point appears to be less important. This may be because multiple pyruvate oxidation systems are present (listed in Fig 6), including pyruvate carboxylase, homologous to the preferred anaplerotic enzyme in another saccharolytic haloarchaeal species, *Haloferax volcanii* [88]. None of the genes encoding pyruvate oxidation systems were differentially expressed in *Δ trmB$_{Har}$* (S8 Fig), though TrmB$_{Har}$ robustly binds near *ppsA* and *porAB*. Additional experiments could reveal whether these genes are differentially expressed in *Har. hispanica* at other points in the growth curve, or whether these binding sites have lost their regulatory function over the course of evolution [89, 90].

The *Har. hispanica* genome encodes two GAPDH homologs belonging to the bacterial type I and archaeal type II clades (*gapI* and *gapII*, respectively). In Archaea, *gapI* homologs occur almost exclusively in saccharolytic haloarchaea and are derived from an ancient horizontal transfer of *gapI* from bacteria [34]. This acquisition, along with an amphibolic phosphoglycerate kinase (*pgk*), grants an additional ATP generation during glycolysis relative to traditional archaeal spED pathways [5]. In *Haloferax volcanii*, *gapII* and *gapI* catalyze the gluconeogenic and glycolytic reactions, respectively [34]. Our data suggest that the functional specificity of GAPDH homologs is preserved in *Har. hispanica*: *gapII* is a direct target of TrmB and its expression is strongly induced in gluconeogenic conditions. In contrast, *gapI* and cotranscribed *pgk*, are repressed in a manner consistent with TrmB-dependent regulation, though it did not pass our significance threshold (p-value = 0.079, S8 Fig, S1 File). In *Har. hispanica*, TrmB-dependent activation of *gapII* is necessary for growth under gluconeogenic conditions (Figs 3 and 6), perhaps replacing regulation of *ppsA*/*pykA* as a critical metabolic control point.

Phenotype data also emphasize *gapII* as a key site of regulation: glycerol, sucrose, and fructose intermediates are predicted to enter gluconeogenesis after reactions catalyzed by GAPDHII (Fig 6C), and the addition of these carbon sources to the medium can abrogate the growth defect in *Δ trmB$_{Har}$* background (Fig 3B). The *Har. hispanica* genome encodes a putative glucose isomerase, HAH_0464, and sucrose hydrolase, HAH_2053, which could enable the conversion of fructose and sucrose to glucose, respectively. *Har. hispanica* also encodes homologs of the bacterial-type phosphoenolpyruvate-dependent phosphotransferase system involved in fructose degradation in *Haloferax volcanii* [11, 16, 39]. These enzymes may contribute to the observed recovery of growth in the presence of sucrose and fructose, but further experiments are needed to test these predictions. In contrast, supplemental pyruvate has no effect on the growth of the deletion strain, indicating that the only way for *Har. hispanica* to generate necessary glucose from pyruvate is via the reverse EMP pathway. Interestingly, ribose has no effect on growth rate in the deletion strain, while xylose partially restores growth, suggesting *Har. hispanica* may be able to generate upper EMP intermediates from xylose but not ribose. This observation is consistent with work showing that *Har. hispanica* uses distinct enzymes to metabolize ribose and xylose and that xylose catabolism utilizes a promiscuous xylonate/gluconate dehydratase [91].

Transcriptome profiling revealed robust TrmB-dependent induction of the tryptophan bio-synthesis operons, including the multi-functional fructose bisphosphate aldolase, HAH_2396. This enzyme is predicted to function in a modified shikimate synthesis pathway as well as cata-lyze the interconversion between GAP and DHAP in the EMP pathway [92]. It is evolution-arily distinct from the archaeal class I fructose bisphosphate aldolase regulated by TrmB in *Hbt. salinarum*. In *Har. hispanica*, there are no binding sites or motif occurrences in the geno-mic regions surrounding the tryptophan biosynthesis operons, indicating additional factors may be involved in the regulation of these promoters. A potential candidate is TrmB homolog HAH_0923, as it exhibits similar TrmB$_{Har}$-dependent expression in response to glucose as the tryptophan operons. Unlike TrmB$_{Hbt}$, which binds near five transcriptional regulators includ-ing itself [10], HAH_3039 and HAH_0923 are the only regulators distinguished as possible tar-gets of TrmB in *Har. hispanica* with binding and expression data, respectively.

Due to the cryptic presence of vector sequence in the *Δ trmB$_{Har}$* strains studied here, we focus mainly on the direct targets of TrmB$_{Har}$ as this interaction term is better able to control for any nonspecific effects of *pyrF* expression and uracil prototrophy. However, we note that the expression of other metabolic pathways is altered in our data (S7 Fig). Specifically, genes involved in the catabolism of ribose, xylose, and arabinose are induced in *Δ trmB$_{Har}$* exclu-sively when glucose is absent, a pattern consistent with indirect repression via as-yet-unknown regulators or mechanisms [91]. Expression of the operon encoding enzymes for the methylas-partate cycle is also elevated in the absence of glucose in *Δ trmB* samples, indicating that TrmB may indirectly repress other major anaplerotic pathways in *Har. hispanica*. Notably, the methylaspartate cycle in haloarchaea is correlated with the ability to synthesize the industri-ally-relevant biopolymer PHBV under favorable environmental conditions [93]. However, we did not observe accumulation of PHBV granules in our media conditions or any significant differential expression of the operon encoding the PHBV pathway (S8 Fig).

## Conclusion

The analyses reported here indicate that TrmB directly activates the expression of genes involved primarily in gluconeogenesis and indirectly regulates tryptophan synthesis in *Har. hispanica*, enabling cells to survive in gluconeogenic conditions. TrmB$_{Har}$ does not directly repress the expression of glycolytic enzymes or other pathways such as cofactor biosynthesis and purine biosynthesis, or acts as a global transcriptional regulator similar to homologs from other archaea [10, 17]. Instead, TrmB$_{Har}$ solely represses the expression of a putative glucose transporter when glucose is absent. Further work is needed to determine if this streamlined regulon in *Har. hispanica* is indicative of sub-functionalization and whether other TrmB homologs present may regulate some of these peripheral functions. Gluconeo-genesis is an essential cellular function in haloarchaea, but it remains to be discovered whether the metabolic fate of glucose made via gluconeogenesis is conserved in metaboli-cally distinct groups.

## Supporting information

**S1 Fig. Genotype confirmation by whole-genome sequencing.** A: Summary of variants iden-tified in each strain. "X" indicates that the mutation (rows) was detected in a given strain (col-umns). Strain designations are given in S1 Table. Representative coverage plots confirming chromosomal deletions for (B) *pyrF* locus for all strains and (C) *trmB$_{Har}$* strains. X-axis pro-vides the genome coordinates. Tables report local read depth, or sequencing coverage, for each strain. D: Confirmation of C-terminal *trmB*-hemagglutinin fusion used for

immunoprecipitation experiments.
(TIF)

**S2 Fig. *Δ trmB* growth.** A: In-trans complementation of *Δ trmB$_{Har}$* in both AKS133 and AKS319 backgrounds. Log-transformed, fitted growth curves of complementation strains and strains harboring the empty vector (EV) grown in the presence or absence of glucose. Shaded regions depict the 95% confidence intervals. B: Area under the growth curve (AUC) of (A), with FDR-corrected significance scores. C: Parent strain and *Δ trmB$_{Har}$* growth in each condition relative to no carbon, measured by AUC. All growth experiments were done with a minimum of 3 biological replicates, each in technical triplicate. Error bars depict the standard deviation of the mean.
(EPS)

**S3 Fig. Relative position of genes, consensus peaks, and motifs for 15 regions identified by ChIP-seq.** Per base coverage of a representative IP sample in the absence of glucose shown in black, an IP sample in glucose is shown in blue. Pile-up was calculated from unextended bam files. The relative location of consensus peaks and motifs is shown below. Magenta bars indicate whether nearby genes were differentially expressed: magenta bars above the line represent genes up-regulated in the *Δ trmB$_{Har}$* mutant. The height of the magenta bar represents the magnitude of change, according to the log fold change (LFC) scale bar to the right of each panel. Gene strand orientation and labels are shown in grey. Motif locations are indicated by the vertical blue line within each panel. Numbers above each panel indicate the genomic coordinates and chromosomal element of the region displayed.
(EPS)

**S4 Fig. Effect of background model on discovered motifs.** Sequences corresponding to peaks were extracted and submitted to XTREME as described in the methods. Yellow highlights indicate the motif reported in Fig 4.
(TIF)

**S5 Fig. *pyrF* transcripts correspond to uracil prototrophy in AKS133.** A: Diagram depicting the process of isolating uracil prototrophic strains from AKS133. B: Average number of transcripts mapping to *pyrF* in AKS133 RNA-seq samples. Standard deviation in parentheses. The point shape indicates the flow cell, or batch, on which the samples were sequenced. C: Log-transformed growth curves in the presence and absence of supplemental uracil and 5-FOA or no uracil. Shaded regions depict the 95% confidence intervals. Inset shows growth curves for individual cultures. Isolates AKS265–71 were obtained using the strategy summarized in A. D: Prototrophic isolates in C were streaked from freezer stock for genomic DNA extraction. Amplification by PCR indicates *pyrF* sequence is present in the genome (left), but that the endogenous deletion is intact (right).
(TIF)

**S6 Fig. *pyrF* expression has negligible impact on *Δ trmB$_{Har}$* fitness and transcriptome.** A: Fitted, log-transformed growth curves showing that AKS319 phenocopies AKS133, and (B) that there is no significant difference between AKS133 and AKS319 in 25 mM glucose as measured by the area under the curve (AUC). Strain colors are preserved in A and B. ** *p*-value < 0.01; **** < 0.0001. C: No significant differences in the growth rate of AKS319 cultures prior to RNA extraction between replicates exhibiting *pyrF* expression and not. Optical density measurements of the cultures harvested for RNA-seq are shown, with corresponding *pyrF* counts summarized in the table below. D: Average counts per transcript are highly correlated across AKS319 samples regardless of *pyrF* expression for both -glucose (N = 2) and

+glucose conditions (N = 2). Average counts per transcript are highly correlated across AKS319 and AKS133 regardless of *pyrF* expression for both -glucose (N = 6) and +glucose conditions (N = 8). Average *pyrF* counts for each comparison are indicated in orange. Data are normalized relative to library size but have not been batch corrected.
(TIF)

**S7 Fig. Strand-specific expression data reveal novel transcript features.** Top row: Novel transcripts observed antisense to HAH_2649 (arCOG11826) and HAH_0885 (SOS response associated peptidase). Bottom row: Targets predicted to encode small proteins appear to have extended 3' UTRs. Green tracks show the per base coverage of transcripts originating from the top strand for representative sample. Blue histograms show transcripts originating from the bottom strand.
(EPS)

**S8 Fig. Expression of genes involved in central carbon metabolism and other catabolic pathways.** A: Normalized expression pattern of genes listed in Fig 6. Yellow indicates elevated expression relative to other conditions, and purple indicates reduced expression. B: Normalized expression of genes involved in characterized anaplerotic and catabolic pathways in *Har. hispanica* [13, 47, 48, 91, 93] and TrmB homologs in Table 1.
(EPS)

**S1 File. Supporting File 1.**
(XLSX)

**S1 Table. Strains used in this study.**
(DOCX)

**S2 Table. Plasmids used in this study.**
(DOCX)

**S3 Table. Primers used in this study.**
(DOCX)

**S4 Table. Motif position relative to predicted transcription initiation elements for direct targets of TrmB$_{Har}$.** Motif sequences are highlighted in purple. Motif occurrences on opposite strands were considered distinct. Darker purple color indicates motif instances on opposite strands overlap. Start codons are highlighted in grey. Putative initiation elements are bolded and underlined (TATA-box and BRE). Other haloarchaea have been reported to frequently lack identifiable TATA sequences [94]. If a promoter element could not be identified, the expected location (i.e., -26/-27 for TATA and -33/-34 for BRE) was underlined.
(TIF)

## Acknowledgments

*Har. hispanica* ATCC33960 and DF60 strains and the pHar plasmid were generously shared by the Xiang research group at the State Key Laboratory of Microbial Resources, Institute of Microbiology, Chinese Academy of Sciences, Beijing, China. We thank Jake Herb for his assistance in generating some of the plasmids and strains reported here, Andrew Soborowski for the list of putative transcriptional regulators in *Har. hispanica*, and Dr. Nicolas Devos and the Center for Genomic and Computational Biology at Duke University for excellent technical assistance. We acknowledge the valuable advice, comments, and feedback of current and former members of the Schmid laboratory during all stages of this project, particularly Saaz Sakrikar, Mar Martinez Pastor, and Sungmin Hwang.

## Author Contributions

**Conceptualization:** Rylee K. Hackley, Cynthia L. Darnell, Amy K. Schmid.

**Data curation:** Rylee K. Hackley, Angie Vreugdenhil-Hayslette, Cynthia L. Darnell, Amy K. Schmid.

**Formal analysis:** Rylee K. Hackley, Angie Vreugdenhil-Hayslette.

**Funding acquisition:** Amy K. Schmid.

**Investigation:** Rylee K. Hackley, Angie Vreugdenhil-Hayslette, Cynthia L. Darnell.

**Methodology:** Angie Vreugdenhil-Hayslette, Cynthia L. Darnell, Amy K. Schmid.

**Project administration:** Amy K. Schmid.

**Resources:** Amy K. Schmid.

**Software:** Rylee K. Hackley.

**Supervision:** Cynthia L. Darnell, Amy K. Schmid.

**Validation:** Rylee K. Hackley.

**Visualization:** Rylee K. Hackley.

**Writing – original draft:** Rylee K. Hackley.

**Writing – review & editing:** Rylee K. Hackley, Amy K. Schmid.

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
