## [Decision Letter · Decision Letter 0]

25 Oct 2023

Dear Dr Schmid,

Thank you very much for submitting your Research Article entitled 'A conserved transcription factor controls gluconeogenesis via distinct targets in hypersaline-adapted archaea with diverse metabolic capabilities' to PLOS Genetics.

The manuscript was fully evaluated at the editorial level and by independent peer reviewers. The reviewers appreciated the attention to an important topic but identified some concerns that we ask you address in a revised manuscript.

We therefore ask you to modify the manuscript according to the review recommendations. Your revisions should address the specific points made by each reviewer.

Yours sincerely,

Sonja Albers

Guest Editor

PLOS Genetics

Lotte Søgaard-Andersen

Section Editor

PLOS Genetics

Reviewer's Responses to Questions

**Comments to the Authors:**

Reviewer #1: Hackley et al characterise the function of the transcription regulator TrmB in the saccharolytic haloarchaeon Haloarcula hispanica using ChIP-seq to determine TrmB binding sites combined with RNA-seq and characterisation of growth phenotypes of a trmB deletion strain. The study is designed to compare the role of TrmB in H. hispanica to its homologue in Halobacterium salinarum that uses glycolysis pathways solely for gluconeogenesis and provide inside into the evolution of a transcription regulatory network. The authors show a more limited role of TrmB in H. hispanica where type II GAPDH and a fructose bisphosphatase (along sugar transporters) appear to be the key enzymes involved in gluconeogenesis via EMP that are regulated by TrmB transcriptionally. The experiments are carefully carried out and interpreted and besides one possible small issue with the data analysis I have only minor comments.

Major:

Figure S3 is meant to visualise the ChIP-seq DNA fragment coverage. For a single binding sites of a transcription factor the ChIP-seq data would be expected to form a single peak over the binding site. Instead, Figure 3 shows consistently double peaks flanking the proposed binding site motif. This very much looks like read coverage (rather than DNA fragment coverage) where reads mapping on the + strand will pile up left of the binding site and reads mapping to the – strand on the right. The sequencing reads might not have been properly extended to the corresponding fragment size, or the estimated average DNA fragment size (150 bp) is incorrect and too short. The authors should revisit the code and ensure that this did not effect any other part of the data analysis.

Minor:

Line 12: “Instead…” maybe turn into “Instead of allosteric regulation, ..”, as it does not follow on directly from the sentence before (presence of novel allosteric regulation points in archaea), but rather the sentence before that (absence of allosteric regulation of FBPA and PFK) .

Line 23/Figure 1: While the statement that TrmB TFs are more abundant in haloarchaea is most likely true, the number of identified TrmB homologues should be stated alongside the number of genomes for each taxonomic group included in the search to provide a fair comparison because of obvious biases in the composition of the data base.

Line 38, line 53, line 479 and elsewhere: pykA, ppsA, gapI, porAB: it might be better to provide the gene names directly next to the full name of the enzymes for clarity.

Line 67: PHBV should be spelled out.

Line 85: “archaeal-type GAPDH”: the manuscript mentions several times different types of GAPDH, it might be worth having a sentence in the introduction that first clarifies the existence of different GAPDH types in archaea.

L. 189: extending the peaks to 300 bp intervals seems somewhat excessive as the real binding site is certainly found in a much narrower range (as evident in the motif position in the intervals), a single bp overlap to merge peaks appears to be too generous under these conditions as it would possibly merge peaks with their summits being possibly ~300 bp apart from each other.

Line 197: “extended to their fragment size”: This should be the estimated average fragment size because the true fragment size remains unknown for single-read NGS data. What does the value of 150 bp reflect? Is this an average across different ChIP-seq libraries.

Line 220: “required st(r)and specific read orientation” The difference between the HiSeq data for ChIP-seq and the NovaSeq data for RNA-seq should be that bowtie should be used in single-read or paired-end mode, respectively. Both mapping data would be strand-specific.

Line 224: “Logarithmic change” should be log2-fold change, naming the base.

Table 2, page 15: I find the RPKM values not informative. Rather some enrichment value relative to the local surrounding region (as in MACS2 peak caller output) or normalisation against the input would be more meaningful.

Page 16, Fig. 3 legend: Some details on the SEA p-value would be helpful. SEA measures enrichment of a motif in a data set, rather than comparing two motifs. Also, in the main text TomTom is cited as method used to compare the two motifs (page 19).

Line 348: As stated above, the intervals of 300 bp seem to be a bit excessive and unnecessarily underestimates the observed enrichment of binding sites in intergenic regions. Narrower, more realistic intervals or indeed looking simply at enrichment of the corresponding motifs themselves in intergenic regions would in my opinion justified.

Line 367: Given that the motif is in part palindromic, I am not sure it is correct to count motif occurrences on different strands at the same position twice.

Line 400: I do not understand this interpretation for the pyrF integration elsewhere that TrmBHar “may be essential”. It is not essential because you deleted it successfully…

Line 434: The proper paper for the BRE to cite would be Bell, Kosa, Sigler, Jackson 1999 PNAS

Line 434-436: It would be fair to say that there seem to be also some deviations from the pattern of TrmB motif position and activation/repression of transcription in the examples of Table S4 (e.g in HAH_0188, binding site overlapping with the BRE, transcription activated)

Line 445: For an uncharacterised sRNA, I would not state that it is monocistronic without any further evidence..

Line 446: Which peak is between two convergent genes? Is this the peak at rank 4 in table 2? Arrows pointing towards each other in the “genomic context” column would be helpful.

Line 459: “has fewer binding sites”: maybe better “’we detected fewer binding sites” given that two different methods (ChIP-chip vs ChIP-seq) and different analysis pipelines were used, for two different organisms… at least to some extend the difference in binding sites could be shaped by the methods. The authors for example for example state that 58 genes close to 67 potential TrmB binding motif occurences are significantly enriched in carbohydrate transport and metabolism (Line 366). If this is true and not solely reflects the TrmB binding sites experimentally identified in ChIP-seq, then there could be additional TrmB binding sites that were beyond the sensitivity of ChIP-seq experiment…

Line 493: It would be good to state the adjusted p-value for gapI and pgk to show the evidence for differential expression (even if not passing the significance threshold.)

Figure S5 legend panel B: What is the meaning of the triangles and dots

Supplementary Tables 1 and 3 are switched around

Figure 2A: Multiple testing correction should be applied

Figure 3C: The y-axis should be -150 to +150 relative to peak position

Figure 5C: The fbp gene is activated by TrmB, the purple colour of the arrow indicating transcription repression is wrong.

Typos/wording:

Line 220 strand -> strand

Lines 381 and 386 pryF

Line 418: “TrmB-mediated induction” -> transcription activation?

Reviewer #2: The manuscript by Hackley et al describes a genomics and genetics study of the conserved TrmB transcription factor, and the role it plays in the saccharolytic haloarchaeal species Haloarcula hispanica. The study has been designed and conducted rigourously, and the data has been analysed and presented clearly. In my opinion, this manuscript requires only a few minor revisions before it is suitable for publication in PLoS Genetics.

General point: when reading the introduction, specifically the description of the glycolysis and gluconeogenesis pathways in line 44 onwards, I found myself searching for an illustration of these metabolic pathways. Just such an illustration then appears as Figure 5, but I feel that this is too late in the manuscript to be of help to a non-specialist. Consider moving Figure 5 (or at least the relevant pathways in Figure 5) to the introduction as Figure 1.

Data query: the authors find that the pyrF-marked deletion vector has repeatedly and independently integrated into the host genome by homologous recombination (lines 381-400). Would this not be evident in the whole genome sequencing data, given that the vector backbone should now be present? Do these sites of integration have anything in common? This information could be used to prevent integration by homologous recombination (e.g. by removing these sequences from the plasmid), which would be of interest to other Haloarcula genetics researchers.

Literature: in lines 350-352 it is stated that archaeal core promoter consensus sequences are still ill-defined, but in lines 432-434 the archaeal promoter architecture is described, with appropriate references. Consider moving this information and citation to lines 350-352.

Grammatical error: throughout the manuscript, the term 'struck' is used to describe the streaking of cells from frozen stocks. The past participle of 'streak' is 'streaked', whereas 'struck' is the past participle of 'strike'. I appreciate that the term 'struck' is used widely in US laboratories, but it is not grammatically correct and risks confusing non-native English speakers.

Typographical error: pyrF is occasionally misspelled as 'pryF', for example on line 381.

**Have all data underlying the figures and results presented in the manuscript been provided?**

Reviewer #1: Yes

Reviewer #2: Yes

PLOS authors have the option to publish the peer review history of their article (what does this mean?). If published, this will include your full peer review and any attached files.

Reviewer #1: **Yes: **Fabian Blombach

Reviewer #2: **Yes: **Thorsten Allers

---

## [Editor Report · Decision Letter 1]

22 Dec 2023

Dear Dr Schmid,

We are pleased to inform you that your manuscript entitled "A conserved transcription factor controls gluconeogenesis via distinct targets in hypersaline-adapted archaea with diverse metabolic capabilities" has been editorially accepted for publication in PLOS Genetics. Congratulations!

Yours sincerely,

Sonja Albers

Guest Editor

PLOS Genetics

Lotte Søgaard-Andersen

Section Editor

PLOS Genetics

Comments from the reviewers (if applicable):

**Data Deposition**

http://datadryad.org/submit?journalID=pgenetics&manu=PGENETICS-D-23-00973R1

**Press Queries**

---

## [Editor Report · Acceptance letter]

8 Jan 2024

PGENETICS-D-23-00973R1 

A conserved transcription factor controls gluconeogenesis via distinct targets in hypersaline-adapted archaea with diverse metabolic capabilities 

Dear Dr Schmid, 

We are pleased to inform you that your manuscript entitled "A conserved transcription factor controls gluconeogenesis via distinct targets in hypersaline-adapted archaea with diverse metabolic capabilities" has been formally accepted for publication in PLOS Genetics! Your manuscript is now with our production department and you will be notified of the publication date in due course.

With kind regards,

Lilla Horvath

PLOS Genetics

On behalf of:
